# Socio-Economic Factors Influencing the Use of Dietary Supplements by Schoolchildren from Małopolska Voivodship (Southern Poland)

**DOI:** 10.3390/ijerph19137826

**Published:** 2022-06-26

**Authors:** Beata Piórecka, Karolina Koczur, Radosław Cichocki, Paweł Jagielski, Paweł Kawalec

**Affiliations:** Department of Nutrition and Drug Research, Institute of Public Health, Faculty of Health Sciences, Jagiellonian University Medical College, 31-066 Krakow, Poland; karolina.koczur@uj.edu.pl (K.K.); radecc10@wp.pl (R.C.); paweljan.jagielski@uj.edu.pl (P.J.); pawel.kawalec@uj.edu.pl (P.K.)

**Keywords:** dietary supplements, eating behaviour, nutritional status, schoolchildren

## Abstract

The use of regular supplementation may be important in alleviating the potential effects of specific nutrient deficiencies. The aim of this cross-sectional study was to assess the socio-economic and lifestyle factors influencing the administration of dietary supplements to schoolchildren from the Małopolskie voivodship. The study was conducted in March–June 2018 on 332 healthy children and adolescents (187 boys, 145 girls) aged 7–14 from the city and municipality of Niepołomice and the city of Kraków. The mean age of the subjects was 10.35 + 1.64 years. In order to assess their diet, a questionnaire was completed, by the parents or the child, on the frequency of consumption of specific products and foods (Food Frequency Questionnaire, FFQ) with added questions on the supplements provided. In assessing nutritional status, basic anthropometric measurements were taken and the BMI index was analysed. To check which factors influenced the use of supplements among respondents, the odds ratio (OR) was calculated. Approximately one-third of the total number of subjects (33.8%) took dietary supplements, most often supplements containing vitamins D and C, followed by multivitamin supplements and omega-3 fatty acids. The least common supplements contained calcium and iron. Dietary supplement intake was significantly higher among children living in rural areas compared to city areas (39.3% vs. 26.5% of respondents; *p* = 0.0150), and among boys compared to girls (37.3% vs. 27.8%; *p* = 0.048). It was observed that children more often received dietary supplements in multigenerational families and in families where at least one parent did not work. This is related to the place of residence of the respondent. Awareness of the need for, and the safe use of, dietary supplements is necessary among parents of children and adolescents.

## 1. Introduction

A varied and balanced diet serves to supply the human body with all the nutrients it needs to function properly. The use of regular supplementation can assist in alleviating the potential effects of deficiency or in supporting the treatment of chronic diseases [1]. It is estimated that over 80% of the world’s population uses dietary supplements or herbal preparations [2]. The largest number of dietary supplements is consumed in the United States of America, while in Europe it is Italy, Russia, and Germany that lead the way [3]. Dietary supplements are consumed by almost all age groups. It has been shown that 72% of adults in Poland report taking dietary supplements, while 48% report using them regularly [4]. In the United States, an average of one-third of children and adolescents take dietary supplements [5]. In the EU, food supplements are regulated as foods and their use is not intended to treat or prevent diseases in humans or to modify physiological functions [6].

In every instance, the use of supplementation should be individualised, supported by the results of diet assessment and nutritional status assessment conducted and in accordance with the recommended doses, especially when it concerns children and adolescents. The main motivation for parents to include supplementation in children is to counter the risk of deficiencies and concerns about the amount of minerals and vitamins ingested in the daily ration [7].

Both vitamin deficiency and excessive vitamin intake cause a number of adverse health effects on the body [8]. As can be concluded from studies conducted in Europe, the intake of vitamins and minerals with the average diet is adequate. However, children and adolescents may be at risk of specific nutritional deficiencies. This mainly concerns vitamin D, iron, and iodine [9]. Current recommendations for the use of supplementation in Poland concern vitamin D, in which it has been determined that 90% of the population is deficient. If insolation guidelines are not fulfilled, vitamin D should be administered at 600–1000 UI/day for children under 10 years of age and 800–2000 UI/day for adolescents and adults throughout the year [10].

The use of supplementation, as a part of an assessment of eating habits, should take into account the influence of a wide variety of socio-economic and lifestyle factors. Higher diet quality has been linked to better socio-economic status (SES). While deciding whether to use supplementation, the following may be important factors: nutritional status, quality of the baseline diet and eating habits, physical activity, and the use of medications associated with chronic disease. 

The aim of the study was to assess the socio-economic and lifestyle factors that influence the administration of dietary supplements in a group of children and adolescents from the Małopolskie voivodship.

## 2. Materials and Methods

### 2.1. Study Design 

This observational cross-sectional study was conducted in March–June 2018 on children from the town and commune of Niepołomice and the city of Kraków. A total of 332 healthy children aged 7–14 years voluntarily participated in the survey conducted at public primary and secondary schools. The exclusion criteria were as follows: (1) taking oral medication for a chronic condition; (2) lack of consent of parents and/or child to participate in the study. The study was conducted in accordance with the principles of medical research ethics contained in the Helsinki Declaration. The research protocol was previously approved by the Bioethics Committee of Jagiellonian University (122.6120.320.2016). We also received the consent of the Probation Officer of Małopolska for the implementation of the study in schools.

### 2.2. Data Collection

The nutritional survey was conducted using the quantitative Food Frequency Questionnaire (FFQ) to assess the consumption of selected foods and meals, based on the Dietary Habits and Nutrition Beliefs Questionnaire, designed by the Behavioral Nutrition Team, Committee of Human Nutrition, Polish Academy of Sciences, validated and recommended for research in the Polish population [11]. Questions were added about the supplementation used and the family’s socio-economic situation and chosen lifestyle factors. The questionnaire was completed by parents of younger children (7–10 years) or independently by adolescents.

In the assessment of nutritional status, selected anthropometric measurements of respondents were taken (height and weight, waist, and hip circumference). To assess nutritional status, body mass index (BMI) was used, which was interpreted according to the WHO criteria (underweight: <10 percentile; normal body weight: 10–85 percentile; overweight: 85–95 percentile, obesity: >95 percentile) in relation to current national percentile charts, taking into account the age and gender of the subjects [12].

### 2.3. Statistical Analysis

In order to check the factors influencing the choice of dietary supplements in the study group, the odds ratio (OR) was calculated by grouping potential socio-economic and lifestyle factors that could influence the intake of the group. These included: gender and age of the child, nutritional status, intensity and frequency of exercise outside school, time spent in front of a TV or computer during the day, diet, and inhalation drugs taken in connection with a chronic disease. Socio-demographic factors also included: mother’s and father’s education and occupational status, place of residence, size of family, whether the family was complete, and whether the family was multigenerational.

The results of anthropometric measurements were presented as the mean value of ±SD and interpretations of the BMI index were used in the evaluation of the nutritional status of the subjects. Other variables were presented as a percentage of responses. Data analyses were performed using PS IMAGO PRO 6 software (IBM SPSS Statistics 26). The level of statistical significance in the assessment was α < 0.05.

## 3. Results

The study involved 187 boys (56.3%) and 145 girls (43.7%). In total, 180 (54.9%) of the respondents lived in cities, while 148 (45.1%) lived in the countryside. The mean age of the children studied was 10.35 + 1.64 years. The socio-demographic situation of the study group is shown in Table 1. Of the respondents, 81 children (24.5%) were raised in incomplete families. For the majority of those participating in the study, parents declared a higher education. In most families (78.1%), both parents worked (Table 1).

Table 2 presents the mean values of anthropometric measurements and interpretations of the BMI index for all those studied, and according to the place of residence. In the group of children living in the countryside, statistically significantly higher mean values of body mass, BMI index, and waist and hip circumference were observed.

The percentage of children declaring participation in P.E. classes at school was high, and no differences were noted between children with respect to the place of residence (children from rural areas—97.14%, children from cities—98.59%, *p* = 0.383). In addition, no differences were observed in the time spent watching TV, taking into account the place of residence.

The study also assessed selected nutritional behaviour of children and adolescents. Among the adolescents studied, first breakfast was consumed daily by 90.5%, second breakfast by 86.2%, lunch by 99.7%, afternoon snack by 55.4%, and dinner by 99.1%. Only 43.8% ate five meals daily. Fruit was consumed once a day or a few times a day by more than half of the respondents, while vegetables, both raw and cooked, were most often consumed a few times a week. Consumption of sweets once or a few times a day was declared by more than half of the respondents (51.9%). No statistically significant difference was observed in the answers of the respondents with regard to the place of residence or gender; therefore, the frequency of consumption of selected food products was presented for the entire study population (Table 3).

Of the total number of respondents, 111 children declared taking dietary supplements, i.e., 33.8% of all respondents. Among children from rural areas, 70 respondents (39.3%) used dietary supplements, whereas in cities the number amounts to 39 respondents (26.53%; *p* = 0.0150). In the total study group, vitamin D (67.9%), vitamin C (46.4%), multivitamin preparations (20.5%), and omega-3 acids (19.1%) were indicated as the supplements most frequently taken. Figure 1 presents the dietary supplements used by children, taking into account the place of residence.

Table 4 presents the percentage of children consuming or not consuming dietary supplements in relation to selected factors. It was shown that boys are more likely to consume dietary supplements than girls (*p* = 0.048), which is statistically significant. Among the users of dietary supplements, boys constituted 37.3% and girls 27.8%. Supplements were also given significantly more often to children living in rural areas and to children living in multi-generational families. The occupational status of parents was also important in the administration of supplements to children. The frequency of supplement intake by children who had both parents professionally active was 30.2%, whereas when one or both parents did not work, the frequency was significantly higher, at 50%.

Table 5 presents only the significant odds ratio results for all variables presented in Table 4. In the study group, potential risk factors which have an impact on taking supplements by children were as follows: place of residence, multi-generation family, and parent’s occupational status. Children from families where both of the parents work had about 60% less chance of taking supplements than children where one or none of the parents were working.

## 4. Discussion

The results of the study indicate that in this group of healthy children aged 7–14 years from southern Poland, only a third of respondents (33.8%) take dietary supplements.

In a study by Bylinowska et al. (2012), conducted between 2005 and 2009 on caregivers of 743 children attending primary schools from the Mazowieckie, Kujawsko-Pomorskie, Łódzkie, and Wielkopolskie voivodships, a similar trend was observed where 40% of children aged 6–12 years took dietary supplements in the year preceding the study [13]. In contrast, a study conducted from June to September 2017 in Wrocław on caregivers of 383 children aged 3–12 years showed that 54.89% of participants gave their children dietary supplements. 

Parents who gave dietary supplements to their children tended to be more trusting of such products than those who did not. It was confirmed that parents transfer their own behavioural patterns to their children, so those who themselves used dietary supplements also gave them to their children [14].

The significant socio-economic factors which influenced the administration of dietary supplements in a group of school children from the Małopolskie voivodship were: place of residence, parents’ occupational status, and living in a multigenerational family. Dietary supplement intake was significantly higher among boys compared to girls. Other socio-economic and lifestyle factors had no effect on the administration of dietary supplements.

In a study conducted by Sicińska et al. (2019), a higher number of children (39.5%) compared to adolescents (20.3%) used dietary supplements during the last year. In this large group of schoolchildren from central-eastern Poland, the significant predictors of dietary supplement use were different depending on age. In children (≤12 years old), socio-economic status, physical activity levels, BMI, and presence of chronic disease were determinants of dietary supplement use. However, in adolescents, it was gender, residential area, BMI (this trend being opposite to that of children), and health status. Some determinants such as higher mothers’ education, consumption of fortified foods, and declaration of diet modifications overlapped in both groups [15].

In a study of preschool children in Kraków, where as many as 21.7% of parents gave supplements to their children daily, it was observed that a higher intake of dietary supplements was associated with correct eating habits [16]. As part of the 2019 national population study, which involved students from 12 provinces in Poland, skipping main meals, especially breakfasts, and the frequent consumption of sweets over fruit and vegetables were observed among the behaviours favouring the development of obesity [17].

The 2017–2018 National Health and Nutrition Examination Survey (NHANES) estimated that the prevalence of supplement use among 3683 children and adolescents in the USA in the previous 30 days was 34.0%. Supplement use was observed to be higher among girls than boys, and the prevalence of supplement intake was age-dependent, with the highest prevalence recorded amongst children aged 2–5 years (43.3%), followed by those aged 6–11 years (37.5%) [5].

In a study conducted in Poland by Pietruszka et al. (2009) where socio-demographic data were collected on 128 children aged 7–11 years in a primary school in Warsaw, parents were asked about dietary habits, use of dietary supplements in the year preceding the study, and consumption of products enriched with vitamins and/or minerals by children [18]. The use of dietary supplements was declared by 49% of the respondents and the use of fortified products by 78%. The simultaneous use of supplementation and fortified products was reported by 42% of respondents. Only 15% of the children did not use any form of dietary supplementation. The intake of vitamins and minerals from all sources posed a risk of exceeding the upper tolerable intake level (UL) for vitamin C, B6, folic acid, magnesium, iron, and zinc.

The results of a representative study conducted in nine European countries, including Poland [19], showed that patterns of voluntary supplementation differed between countries and were responsible for the largest differences in total nutrient intake. Dietary supplement use was highest in Finland and Denmark. In contrast, supplementation in Poland was mostly within the average with respect to the rest of the countries. The exception was calcium whose supplementation in Poland was one of the lowest. This was also confirmed in the present study in children from the Małopolskie voivodship. In the total study group, the most consumed dietary supplements by school children contained vitamins D and C, multivitamin preparations, and omega-3 fatty acids, and least consumed supplements contained calcium and iron.

The insufficient intake of certain substances during adolescence, such as vitamin D, calcium, or magnesium, may lead to inadequate bone mineralization, which increases the risk of osteoporosis in the future. In a study conducted on 461 girls aged 10–15 years in the Lubelskie voivodship using 24 h dietary interviews, large deficits in dietary intake of vitamin D and calcium were observed [20].

According to guidelines for Central Europe, vitamin D supplementation is recommended as the primary strategy for balancing diet deficiencies in the population [10]. In a study of healthy schoolchildren from the Małopolskie voivodship, among those taking dietary supplements, as many as 67.9% took vitamin D supplements.

A study of Irish adolescents assessed vitamin D intake and its contribution through supplementation and fortified foods as an addition to the staple diet. The study involved 1035 children aged 5–17 years. Use of the vitamin as a dietary supplement was reported by 21% of children aged 5–8, 16% of those aged 9–12, and 15% of adolescents aged 13–17 [21]. In contrast, a Belgian study (VITADEK) assessed the total intake of fat-soluble vitamins from different sources (i.e., regular diet, fortified foods, dietary supplements) among 3200 individuals aged 3–64 years. Among adolescents aged 11–17 years, the proportion of vitamin D supplementation was 41% [22].

In a study of children from Niepołomice and Kraków, it was observed that those who lived with their grandmother or grandfather were significantly more likely to receive dietary supplements. From the literature review, parents’ education was an important factor influencing the administration of supplements to children. Our own study, where more than half the parents declared higher education, did not confirm this.

A study conducted by Yoko et al. (2016) on Japanese parents found that women with higher education were statistically more likely to give dietary supplements to their children [23]. Similarly, in a study by Bylinowska et al. (2012) conducted on Polish schoolchildren, it was observed that women with higher or secondary education were more likely to give dietary supplements to their children [13]. A study conducted by Evans et al. (2012) on 9417 children found that 54% of parents giving supplements to their children had a university degree [24]. Similar results were found in a study by Kozyrska et al. (2010) on parents of children aged 7–12 years in the Mazowieckie voivodship [25]. In this study, 61.7% of mothers who administered dietary supplements to their children had higher education.

A study by Szymelfejnik et al. (2019) conducted on 2258 children from primary schools in the Kujawsko-Pomorskie voivodship showed a similar relationship. This study assessed the use of dietary supplementation among children attending the first grade of primary school, in relation to health status and socio-economic factors. Data were obtained using a questionnaire completed by parents. Supplementation was observed in 32.4% of children, regardless of age and gender. Among parents with a low level of education, the use of dietary supplements by their children was significantly lower. The main reason for the use of dietary supplements in this group was the child’s health status. The intake of dietary supplements was also statistically significantly dependent on the place of residence. The highest intake of dietary supplements was observed in the children from a medium town, then in the group living in the countryside and in a big town, while the lowest intake of dietary supplements concerned children living in small towns [26]. In a study on children from the south of Poland, a similar relationship was observed, where children living in the countryside received dietary supplements more often.

In Poland, for several years now, there has been an alarming increase in the sale of dietary supplements. Their widespread availability and intensive advertising sometimes result in their unjustified use. Supplements are a form of food and are not medicinal products; therefore, they do not contain information about potential health risks on the packaging [27]. The Committee of Human Nutrition Science of the Polish Academy of Sciences, speaking on the use of dietary supplements by adults, draws attention to the fact that dietary supplements that have been introduced to the Polish market are safe, but their improper use may pose a threat. Taking dietary supplements should follow consultation with a dietician, doctor, or pharmacist, as there is a risk of overdose or interaction of ingredients contained in supplements with food, medicines, or other supplements. The use of a varied diet should be considered as the first step in improving nutritional status and health [28]. Current studies show an increasing percentage of Polish adults who declare taking dietary supplements because of the epidemic situation [29].

A limitation in assessing the results of the present study is the lack of information on the dosage of nutrients in the supplements administered to children and whether supplements are taken on the basis of medical recommendations.

The study was conducted on a large group of healthy school children of different ages in southern Poland and assessed the prevalence of supplement use in a region where data on supplement use among children and adolescents have not been updated.

## 5. Conclusions

In the study group, only one-third of school children and adolescents take dietary supplements, and vitamin D is the most frequently indicated. The information strategy currently implemented concerning the role of this vitamin in compensating for dietary deficiencies is insufficient. Awareness of the need for, and the safe use of, dietary supplements is necessary among parents of children and adolescents.

## Figures and Tables

**Figure 1 ijerph-19-07826-f001:**
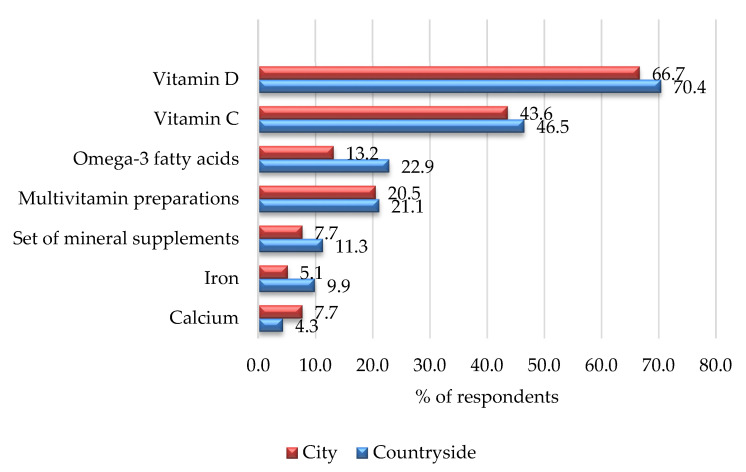
Percentage of children taking supplements by place of residence.

**Table 1 ijerph-19-07826-t001:** Socio-demographic characteristics of the study group.

Selected Socio-Demographic Variables	N	%
**Gender**		
Male	187	56.3
Female	145	43.7
**Age**		
Up to 11 years	213	64.2
Above 11 years	119	35.8
**Place of residence**		
countryside	148	45.1
city	180	54.9
**Family completeness ***		
complete	250	75.5
incomplete	81	24.5
**Education of the mother**		
primary	2	3
vocational	37	11.5
secondary	95	29.4
higher	189	58.5
**Education of the father**		
primary	1	0.3
vocational	70	22.4
secondary	97	31.1
higher	144	46.2
**Occupational status of parents**		
Both parents work	286	78.1
Only the father works	57	18.1
Only the mother works	12	3.85

* Complete family—both parents live with the child.

**Table 2 ijerph-19-07826-t002:** Results of anthropometric measurements and interpretations of the BMI index of children according to place of residence.

Variable		Total (N = 327)		Countryside (N = 180)		City (N = 147)		
		$\bar{x}$ ± SD		$\bar{x}$ ± SD		$\bar{x}$ ± SD		*p*
Height (m) *		1.44 ± 0.12		1.45 ± 0.12		1.42 ± 0.11		0.037
Weight (kg) *		37.87 ± 10.83		39.80 ± 11.65		35.50 ± 9.24		0.001
BMI (kg/m^2^) *		17.98 ± 3.14		18.51 ± 3.36		17.33 ± 2.71		0.001
Waist circumference (cm) *		62.33 ± 8.01		63.82 ± 8.63		60.51 ± 6.78		<0.001
Hip circumference (cm) *		75.86 ± 9.08		77.89 ± 9.47		73.38 ± 7.93		<0.001
Variable	Category	N	%	N	%	N	%	*p*
BMI interpretation *	Underweight	26	7.9	13	7.2	13	8.8	0.0136
	Normal body weight	249	75.5	127	70.6	119	81.0	
	Overweight	33	10.0	26	14.4	7	4.8	
	Obesity	22	6.7	14	7.8	8	5.4	

N—number of children, X—arithmetic mean, SD—standard deviation, *p*—significance level of differences, BMI—body mass index, *—statistically significant difference, *p*—U Mann–Whitney test.

**Table 3 ijerph-19-07826-t003:** Frequency of consumption (% of answer) of selected food products for all the schoolchildren.

Food Product	Never	One to Three Times a Month	Once a Week	A Few Times a Week	Once a Day	A Few Times a Day
white rice, plain pasta or small groats	1.2	8.3	23.8	53.7	12.3	0.6
plain bread	0.9	1.8	1.2	10.1	24.8	61.0
wholemeal bread	29.7	25.3	14.2	16.8	8.9	5.1
buckwheat groats, oatmeal, wholemeal pasta or other coarse groats	22.2	24.9	17.5	25.5	7.7	2.2
yellow cheeses, processed cheeses, mouldy cheeses	10.5	12.0	14.8	38.3	17.0	7.4
cottage cheese	11.2	12.1	24.5	33.9	14.9	3.4
cold cuts, sausages, wieners	2.1	5.5	8.0	44.3	26.6	13.5
fish, fish preparations	11.0	31.0	41.4	13.8	1.3	1.6
so-called white meat dishes	0.9	4.7	15.8	66.1	9.6	2.8
so-called red meat dishes	6.5	20.4	31.5	38.0	3.1	0.6
legume dishes	24.7	48.5	16.7	9.3	0.6	0.3
nuts, seeds	18.6	37.2	24.1	16.4	2.2	1.5
Eggs	3.4	10.1	30.8	48.5	5.5	1.8
butter	4.0	4.0	5.2	17.5	30.7	38.7
Margarines	61.1	8.0	5.1	9.2	8.9	7.6
potatoes (not including chips or crisps)	2.8	4.0	8.9	65.0	16.9	2.5
canned vegetables, marinated or pickled vegetables	22.7	30.4	20.5	22.4	3.1	0.9
raw vegetables	12.1	15.8	12.1	32.8	18.6	8.7
cooked vegetables	11.2	13.7	14.3	43.2	16.8	0.9
fruit	0.3	1.6	6.6	30.6	35.0	25.9
canned meat	0.3	72.9	21.7	2.9	1.3	1.0
sweets	0.9	4.0	9.8	33.3	28.7	23.2
fast food	6.1	67.6	21.1	4.6	0.3	0.3

**Table 4 ijerph-19-07826-t004:** Percentage of children with respect to selected factors divided in relation to whether or not they consume dietary supplements.

Variables	Time	% of Respondents Who Consume Food Supplements	% of Respondents Who Do Not Consume Food Supplements	*p*
Sex	Male	37.3	62.7	0.0480
	Female	27.8	72.2	
Age	7–10	33.2	66.8	0.7326
	11–14	35.0	65.0	
Place of residence	Countryside	39.3	60.7	0.0150
	City	26.5	73.5	
Nutritional status	Underweight and normal	34.6	65.4	0.4848
	Overweight and obese	29.6	70.4	
Multiple family	Not multiple	32.5	67.5	0.3964
	Multiple	37.5	62.5	
Multi-generational family	Two generations	30.3	69.7	0.0255
	Many generations	43.3	56.7	
Mother’s education	Basic, vocational	41.0	59.0	0.3349
	Secondary, higher	33.2	66.8	
Father’s education	Basic, vocational	36.6	63.4	0.7029
	Secondary, higher	34.2	65.8	
Parents’ occupational status	One or both do not work	50.0	50.0	0.0021
	Both work	30.2	69.8	
Frequency and intensity of exercise	Low	33.0	67.0	0.8312
	High	34.2	65.8	
Time spent in front of TV or computer per day	Less than 2 h	32.7	67.3	0.5480
	2 h or more	36.0	64.0	

*p*—chi square test.

**Table 5 ijerph-19-07826-t005:** Potential predictors of use of supplements in studied group.

Variables	OR	95%CI
**Place of residence**		
Countryside	1.0	REF
City	0.557	0.347–0.895
**Multi-generational family**		
Two generations	1.0	REF
Many generations	1.763	1.069–2.908
**Parents’occupational status**		
One or both do not work	1.0	REF
Both work	0.433	0.252–0.744

## Data Availability

Not applicable.

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
