# Peer review of "Socio-Economic Factors Influencing the Use of Dietary Supplements by Schoolchildren from Małopolska Voivodship (Southern Poland)"

_ijerph, 2022, doi:10.3390/ijerph19137826_

Round 1

Reviewer 1 Report

Introduction:

The introduction is written in detail about the importance of Dietary Supplements and their use across the region. However, I think the details related to the market value of dietary supplements are unnecessary in this paper where the authors are not focusing economic analysis of the dietary supplements.

The argument related to Socio-Economic Factors Influencing the Use of Dietary Supplements is totally missing from the introduction. I would suggest adding some literature to build your argument on why is it important to study socio-demographic characteristics that influence the use of dietary supplements? 

Methods
I would suggest dividing the method section is subsections such as study settings, sampling strategy, sample size, tool, variables and measurements inclusion, and exclusion criteria and analysis. 

Some of the components are part of the method section, however, some of the components are totally missing such as sampling strategy, study tool, variables, measurement, etc., 

The sampling strategy is missing--explain which sampling strategy was used for the selection of the participants.

Two sentences from lines 85-87 can be mentioned in the result section.

Details related to anthropometric measurements are missing. How did the authors measure the anthropometric indicators of children what was the cut-off values? 

Results

The authors can also report stunting, wasting, and underweight apart from just presenting the mean and standard deviation in Table 02

How did the authors select some variables in the bivariate model (table -05; page 06). What was the rationale for choosing some variables in this model? Moreover, also discuss this model in the method section.

There are two tables with the name of table 05--Please correct it what is the logic of developing table 05 on page 07 before the discussion section?

Discussion

The discussion section is well written. However, I would suggest focusing on the research questions and omitting some unnecessary arguments in the discussion section. For example, the starting paragraphs of the discussion section are too large that can be divided into some small meaningful paras. And the discussion section on pages 09-10 is small without coherent sequences. 

Conclusions:

The conclusion needs to be rewritten. The conclusion of this paper is very weak and superficial. For example, the authors conclude their argument to increase the awareness of children and adolescents and their parents of the need for and safety of dietary supplements
In the light of the above studies, it is also important to note that awareness should be increased.

Author Response

Dear Reviewer, 

Thank you for your suggestions on improving the manuscript. 

Reviewer 2 Report

The  experimental design is not clear and hence the result section is difficutl to follow.

Lines 48-55 “Current recommendations for the use of supplementation in Poland concern vitamin D, whose deficiency has been determined for 90% of the population. Vitamin D should be administered at 600-1000 UI/day for children under 10 years of age and 800-2000 UI/day for adolescents and adults throughout the year [25].”

How is this recommendation implemented? Is there an official information strategy?

If no official strategy to inform parents has been implemented, we cannot state that this is a recommendation. If on the contrary an official communication strategy is implemented, why only one third of  the total number of subjects (33.8%) takes dietary supplements? Shall we conclude that the information strategy is inefficient?

Lines 222-224 “Taking dietary supplements should be consulted with a dietician, doctor or pharmacist as there is a risk of overdose or interaction of ingredients contained in supplements with food, medicines or other supplements.”

Looking at the Tables, it seems that it was non investigated whether the supplements were taken due to a medical advice or not. Moreover, it is not specified if the amount taken is correct or not.

Lines 91-92: The questionnaire was completed either by parents or independently by adolescents”

This sentence is not clear: it is necessary to define who compiled the questionnaire

Author Response

(The authors gave the same response as above.)

Reviewer 3 Report

In this manuscript, Piorecka et al. aimed to assess the social-economic factors influencing the administration of dietary supplements to a group of children and adolescents from the Malopolskie voivodship. The Food Frequency Questionnaire (FFQ) was used to assess the consumption of selected foods and meals Questions were added about the supplementation used and family’s socio-economic situation. The design and result are scientifically meaningful. With the result, it is reasonable to conclude that it’s important to note that the awareness of children and adolescents and their parents of the need for and safety of dietary supplements should be increased. I have no comments on this manuscript and suggest the editor accept.

Author Response

Dear Reviewer,

Thank you very much for accepting the submitted work. We have been corrected text with the recommendations of the other Reviewers.

Reviewer 4 Report

The authors investigated the socio-economic factors influencing the administration of dietary supplements to schoolchildren from the Malopolskie voivodship.

The authors investigated the socio-economic factors influencing the administration of dietary supplements to schoolchildren from the Malopolskie voivodship. Some concerns relevant to this manuscript are listed below.

The authors should clearly state the significance to conduct this research.

The research methodology is not well defined.

What are the research questions/assumptions?

How was the sample size calculated?

Was the questionnaire validated? 

How to deal with missing data?

Some parts of the methods should be in the results e.g. the number of boys and girls and the mean age.

Author Response

(The authors gave the same response as above.)

Round 2

Reviewer 1 Report

I have reviewed the edits of the authors. They have revised the draft accordingly and please proceed further. 

Reviewer 2 Report

The points raised by this reviewer have been addressed

Reviewer 4 Report

The authors have revised and improved the manuscript according to the reviewer's comments.